# The Interaction between Occupational Stress and Smoking, Alcohol Drinking and BMI on Hypertension in Chinese Petrochemical Workers

**DOI:** 10.3390/ijerph192416932

**Published:** 2022-12-16

**Authors:** Zhihui Gu, Yunting Qu, Hui Wu

**Affiliations:** Department of Social Medicine, School of Health Management, China Medical University, No. 77 Puhe Road, Shenyang North New Area, Shenyang 110122, China

**Keywords:** occupational health, occupational stress, hypertension, alcohol drinking, smoking, BMI

## Abstract

Background: Hypertension is one of the most vital risk factors for cardiovascular diseases, so we wanted to explore the impact of the interaction between occupational stress and smoking, alcohol drinking and BMI on hypertension in Chinese petrochemical workers. Methods: A total of 1488 employees participated. Questionnaires included the value of blood pressure, occupational stress (assessed by the effort–reward imbalance scale), demographic factors and work conditions. Multivariable logistic regression was used to explore related factors, and the interactions between occupational stress and smoking, alcohol drinking and BMI on hypertension were analyzed using an additive model. Results: The prevalence of hypertension was 34.3%. Age ≥ 56 (OR = 3.19, 95%CI: 1.841–5.526), male (OR = 1.436, 95%CI: 1.056–1.954), BMI ≥ 25 (OR = 1.86, 95%CI: 1.468–2.346), smoking (OR = 1.52, 95%CI: 1.191–1.951) and alcohol drinking (OR = 1.53, 95%CI: 1.180–1.972), ERI > 1 (OR = 1.50, 95%CI: 1.133–1.960) are risk factors for hypertension, and a higher education level (OR = 0.57, 95%CI: 0.438–0.754) is a protective factor against hypertension. Positive interactions existed between occupational stress and smoking (RERI = 2.134, AP = 0.328, S = 1.635), alcohol drinking (RERI = 2.332, AP = 0.361, S = 1.746) and BMI (RERI = 1.841, AP = 0.340, S = 1.717) on hypertension in petrochemical workers. Conclusions: Age, gender, educational level, BMI, smoking, alcohol drinking and occupational stress are closely related to the risk of hypertension. There are also positive interactions between occupational stress and alcohol drinking, smoking and BMI, which have a certain impact on hypertension.

## 1. Introduction

Hypertension is the most important modifiable risk factor for cardiovascular disease and mortality, and it has become one of the most common chronic diseases in recent years. By 2025, there will be about 1.56 billion patients with hypertension in the world, accounting for 29% of the global adult population with hypertension [1]. A survey on the prevalence of hypertension among adults in northeast China showed that the crude prevalence of hypertension was 30.8% in 2012 [2]. Hypertension is one of the most vital risk factors for cardiovascular diseases. Stroke, heart failure, myocardial infarction and renal failure are the main complications [3]. Hypertension may affect work ability, job opportunities and family income, reduce the quality of life and directly increase economic costs [4]. Therefore, it is necessary to screen early, identify relevant risk factors and prevent hypertension in advance.

In addition to the known genetic and behavioral factors that contribute to hypertension, occupational stress (or job strain), resulting from a lack of balance between job demands and job control, is considered one of the frequent factors in the etiology of hypertension in modern society [5]. Occupational stress, also known as work stress, refers to the physical and psychological stress caused by work or factors related to work and is the physical or psychological stress response caused by work requirements that are beyond the ability of workers [6]. According to the effort–reward imbalance model, when workers’ perceived job rewards (including career security, salary, respect, promotion opportunities, etc.) do not match their perceived efforts (such as work obligations, work requirements, etc.), this imbalance may cause a state of emotional distress and lead to adverse health outcomes [7]. Occupational stress causes stress reactions in the neuroendocrine system [5], which would lead to unhealthy conditions for workers in petrochemical companies. Petrochemical enterprises play a key role in meeting the growing needs of China’s production economy and are high-risk enterprises. Additionally, the transformation of the industrialization mode has improved the level of mechanical automation, which requires these workers to receive training on new knowledge and new technology after work, bringing additional stress to their physical and mental health. Petrochemical workers are engaged in high-intensity work, with difficult working conditions, shift work, irregular work and rest, harsh conditions such as high temperature, noise and other harmful factors, and high stress on work and family, which easily leads to hypertension [8]. Some scholars have studied and confirmed that chronic exposure to stressors in this population is associated with hypertension [9]. Considering today’s rapid process of modernization, we can speculate that the increase in work stress in petrochemical workers may be one of the reasons for the high prevalence of hypertension [10]. Therefore, occupational stress and hypertension among employees in petrochemical enterprises deserve attention. The differences in people’s specific working conditions, economic income, consumption level, family structure, interpersonal relationship, education level, occupation of leisure time, housing and social services, etc., make obvious differences in the lifestyles of different classes, strata, occupational groups and individuals in the same society. Petrochemical workers have strenuous working conditions, low economic income, low education levels, less leisure time and poor social services, which lead to a poor lifestyle. Therefore, it is very important to study the lifestyles of petrochemical workers. Active smoking is a widely accepted risk factor for hypertension and is recognized as a major public health problem [11]. Alcohol usage is a more frequent contributor to hypertension than is generally appreciated [12]. However, smoking and alcohol usage are very common among petrochemical workers [9]. Body mass index (BMI) has also been shown to be closely associated with the development of hypertension in workers of petroleum enterprises [13]. It is necessary to include lifestyle factors such as smoking, drinking and BMI in research on petrochemical workers.

Therefore, this study evaluated the prevalence of hypertension among workers in petrochemical enterprises in China and analyzed the risk factors affecting the prevalence of hypertension among petrochemical employees and the interaction between these factors. Referring to previous studies, demographic factors (age, gender, marital status, education level, body mass index (BMI), smoking, alcohol drinking, physical exercise and monthly income (CNY)), working conditions (job rank, occupational categories, weekly working hours, work shift and night shift) and occupational stress (effort–reward imbalance: ERI) were included. This study selected petrochemical workers as the research object, explored the risk factors for hypertension in petrochemical workers, and further tested the interaction between occupational stress and smoking, alcohol drinking and body mass index.

## 2. Materials and Methods

### 2.1. Study Design and Data Collection

A cross-sectional survey was conducted in Panjin City, Liaoning Province, China, in May 2019. Liaoning Province is an important heavy-industry base in China, and Panjin City has gathered a lot of petrochemical enterprises. Finally, we randomly selected a large factory, and a total of 1900 workers were willing to participate. After obtaining the informed consent of each participant, a self-administered questionnaire was directly distributed to the participants. Complete responses were obtained from 1488 individuals (effective response rate: 78.81%). The study was approved by the Committee on Human Experimentation of China Medical University, and the study procedures were in accordance with ethical standards. Inclusion criteria: workers who worked in the same position for more than 1 year. Exclusion criteria: workers with a family history of mental illness and/or hypertension; workers with clear secondary hypertension.

### 2.2. Blood Pressure (BP) Measurement and Hypertension Diagnosis

The location for filling in the questionnaire and measuring BP was a quiet office in the factory. Investigators were divided into four groups to carry out the investigation at the same time. The investigation lasted for 10 days, and about 200 people were investigated every day. BP measurement was the most accurate in the morning when the participants had an empty stomach, so it started at 8:00 every morning. Before filling in the questionnaire, trained nurses used BP monitors to measure the BP of workers. Participants were asked to sit for at least 10 min. Systolic blood pressure (SBP) and diastolic blood pressure (DBP) were measured twice in the right arm for each subject. The mean score of the last two measurements was considered the value of BP. If the SBP was ≥140 mmHg, or/and the DBP was ≥90 mmHg, the subject was asked to rest for 15 min, and BP was retaken twice. The mean of the two measurements was the final result. The above measurement method was performed according to the China Chronic Disease Surveillance Project, published by the National Work Group of the project. In our study, 398 workers with SBP ≥140 mmHg and DBP ≥90 mmHg were defined as having hypertension; 56 workers with SBP ≥140 mmHg and 39 workers with DBP ≥90 mmHg were defined as having hypertension; and 18 workers whose value of BP was normal but took antihypertensive drugs for a long time were also classified as hypertension patients by the medical staff according to their symptom descriptions.

### 2.3. Demographic Variables, Lifestyle and Working Characteristics

Demographic characteristics include age, gender, marital status, educational level, BMI, smoking, alcohol drinking, physical exercise status and monthly income (CNY). Body mass index (BMI) was categorized as “<25” or “≥25” according to the evaluation criterion for overweight and obesity, and the “≥25” group was defined as having overweight and obesity. “Smoking” was defined as “Yes” or “No” (“No” means never smoked). “Alcohol drinking” was defined as “Yes” or “No” (“No” means never drinking alcohol). “Physical exercise status” was categorized as follows: never; occasional; and regular (≥3 times per week for 30 min each session).

Working characteristics include job rank, occupational category, weekly work time, work shift and night duty. “Job rank” was categorized as “Head workers” and “Staff workers”. “Occupational category” was categorized as “refinery workers”, “chemical workers”, “transportation workers” and “other workers”. “Work shift” (single-shift system or multi-shift system) was categorized as “Yes” or “No”. “Night shift” (work during the day or work at night) was defined as “Yes” or “No”.

### 2.4. Measurement of Occupational Stress

We used the Chinese version of the effort–reward imbalance (ERI) scale to measure occupational stress [14]. The ERI scale includes extrinsic effort (6 items) and reward (11 items) using a 5-point scoring method, and the higher scores indicate higher efforts and rewards. The effort–reward ratio was calculated by using the predefined formula: ERI = effort/(reward × 0.5454). An ERI score greater than 1 indicated occupational stress. The Cronbach’s coefficients of extrinsic effort and reward scales were 0.784 and 0.886 in the present study.

### 2.5. Statistical Analyses

The chi-square test was used to examine the distribution of demographic factors, working conditions, lifestyle and occupational stress in relation to hypertension, and multivariable logistic regression was used to determine the influencing factors. Finally, we used an additive model to test the additive interactions between occupational stress and smoking, drinking and BMI on the risk of hypertension [15]. The relative excess risk caused by the interaction was calculated (RERI = RR11 − RR10 − RR01 + 1), and RERI = 0 means no interaction or full additivity; RERI > 0 indicates a positive interaction or greater than additive; RERI < 0 indicates a negative interaction or less than additive. The attributable proportion due to the interaction (AP = RERI/RR11) was calculated. AP = 0 means no interaction or exactly additive; AP > 0 means a positive interaction or more than additive; AP < 0 means a negative interaction or less than additive; AP ranges from −1 to +1. The synergy index [S = (RR11-1)/(RR01-1) + (RR10-1)] was calculated. The attribution ratio due to the interaction (AP = RERI/RR11) was calculated. AP = 0 means no interaction or complete additivity; AP > 0 means a positive interaction or greater than additive; AP < 0 indicates a negative interaction or less than additive; AP can range from −1 to +1. The synergy index was calculated by [S = (RR11-1)/(RR01-1) + (RR10-1)]. S = 1 indicates no interaction or complete additivity; S > 1 means a positive interaction or greater than additive; S < 1 indicates a negative interaction or less than additive; S ranges from 0 to infinity. If there were two factors, RR11 was the relative risk of disease [16]. If one factor existed but the other did not, RR10 and RR01 were the relative risks of disease. IBM SPSS Statistics 21.0 (IBM, Asia Analytics Shanghai) was used to conduct the statistical analyses, with a two-tailed probability value of <0.05 considered to be statistically significant.

## 3. Results

### 3.1. Characteristics of the Classified Subjects

Table 1 shows the characteristics of the study subjects. The average age of the participants was 43.22 ± 8.92 years old (female: 45.18 ± 6.53; male: 42.61 ± 9.46). The prevalence of hypertension among petrochemical workers was 34.34%. In the ERI model, workers with a value higher than 1.0 were characterized as having “occupation stress”; according to this, 21.6% of the petrochemical workers had “occupation stress”. The results of univariate analyses between hypertension and all classification variables are shown in Table 1. Age, gender, marital status, educational level, BMI, smoking, alcohol drinking, job rank, occupational category (*p* < 0.01) and weekly work time (*p* < 0.05) were significantly related to hypertension. Among the petrochemical workers, occupational stress was also significantly related to hypertension (*p* < 0.01).

### 3.2. Analyses of Influencing Factors for Prevalence of Hypertension

The results of the multivariable logistic regression of factors associated with hypertension are shown in Table 2. Age ≥ 56 (odds ratio (OR) = 3.19, 95% confidence interval (CI): 1.841–5.526), male (OR = 1.436, 95%CI: 1.056–1.954), BMI ≥ 25 (OR = 1.86, 95%CI: 1.468–2.346), smoking (OR = 1.52, 95%CI: 1.191–1.951), alcohol drinking (OR = 1.53, 95%CI: 1.180–1.972) and ERI > 1 (OR = 1.50, 95%CI: 1.133–1.960) were risk factors for hypertension, and higher education level (OR = 0.57, 95%CI:0.438–0.754) was a protective factor against hypertension.

### 3.3. Calculation of Additive Interaction Index of Occupational Stress and Smoking, Alcohol Drinking and BMI on Hypertension

All of the results of the interaction analysis are shown in Table 3. After adjusting for potential covariates, including age, gender and educational level, the interactions between occupational stress and smoking, alcohol drinking and BMI were significant. The results showed that there would be 2.134 (RERI = 2.134) relative excess risk due to the additive interaction between occupational stress and smoking, and 32.8% (AP = 0.328) of hypertension was caused by the additive interaction between two risk factors and the S index was 1.635. Similarly, there would be 2.332 (RERI = 2.332) relative excess risk due to the additive interaction between occupational stress and alcohol drinking, and 36.1% (AP = 0.361) of hypertension was caused by the additive interaction between two risk factors and the S index was 1.746. There would be 1.841 (RERI = 1.841) relative excess risk due to the additive interaction between occupational stress and BMI, and 34.0% (AP = 0.340) of hypertension was caused by the additive interaction between two risk factors and the S index was 1.717.

## 4. Discussion

In our study, the prevalence of hypertension among Chinese petrochemical workers was 34.34%, which is higher than that in Liaoning Province adults (32.7%) and Chinese adults (27.90%) over 18 years old [17,18]. Compared with other types of work in China, the prevalence of hypertension among Chinese petrochemical workers is still higher than among other workers, such as coal miners (33.7%) and iron and steel workers (25.6%) [19,20].

Our study also indicated that age ≥ 56, male, BMI ≥ 25, lower education, smoking and alcohol drinking were risk factors for hypertension. A previous study has confirmed that the prevalence of hypertension increases with age and was 10.53% in the group aged 15–44 years and 39.68% in the group aged 45–64 years [21]. In our study, the prevalence of hypertension was 25.10% in the group aged 18–45 years and 44.60% in the group aged 46–65 years. The results also showed a higher prevalence of hypertension in men, which is consistent with previous studies [22]. However, the number of women in this study is very small, and conclusions from our data should be drawn with caution. New health problems such as overweight and obesity occur due to the enrichment of material life, and their link to hypertension has been confirmed [23]. Our study also showed that the prevalence of hypertension was 28.74% in the group with BMI < 25 and 43.57% in the group with BMI ≥ 25. This study found that the education level as a protective factor can reduce hypertension (24.65% in the senior high school or above group; 43.57% in the junior college or below group), similar to previous studies [21], which may be related to the type of job. Highly educated workers may conduct intellectual activities with low work intensity; at the same time, they were familiar with daily hypertension prevention knowledge and would pay more attention to their own health [24]. On the contrary, workers with low educational backgrounds were more engaged in manual activities and were not familiar with hypertension protection knowledge, which led to hypertension. Secondly, our results showed that current smoking or past smoking is linked to an increased risk of hypertension (28.92% in the smoking group; 44.86% in the non-smoking group), which is consistent with previous reports. Smoking can increase blood viscosity, stimulate the adrenergic nervous system and accelerate the onset of microvascular and macrovascular diseases [25]. In addition, our study showed that alcohol drinking was also a risk factor for hypertension (26.69% in the alcohol-drinking group; 38.79% in the non-alcohol-drinking group), consistent with previous studies [26]. All of these results suggest that lifestyle changes, such as reducing or quitting smoking and alcohol drinking, eating healthy and exercising more, can help to lower the prevalence of hypertension.

Our study also found that occupational stress was associated with the prevalence of hypertension (32.47% in the ERI ≤ 1 group; 41.12% in the ERI > 1 group), supporting the conclusion that the higher the level of stress is, the higher the risk of hypertension becomes [27]. After controlling for confounding factors, this study further confirmed the conclusion that occupational stress is related to the prevalence of hypertension. According to occupational stress theory, long-term and chronic stress stimulates the body to produce a series of physiological responses and raises blood pressure through neuroendocrine mechanisms [28]. Therefore, it is suggested that enterprise managers reasonably arrange workers’ work tasks and pay attention to workers’ emotions, which can appropriately reduce occupational stress and improve their ability to deal with stress. Our epidemiological data suggest that some working characteristics might be not statistically significant. However, some past studies have shown that the work shift, job rank or occupational category may also affect the occurrence of hypertension [29]. Therefore, we need to analyze the impact of working characteristics on hypertension in further studies.

This study has shown that when occupational stress exists simultaneously with smoking, alcohol drinking or obesity, the risk of hypertension is higher than that of one of these factors (OR11 > OR10 + OR01 − OR00) [30]. It can be seen that occupational stress interacts with smoking, alcohol drinking and obesity to affect hypertension. Increases in smoking, alcohol-drinking behavior and BMI were also related to psychological stress. Due to the comforting effect of nicotine in tobacco, many people in life regard smoking as a way to relieve stress and tension. The research results showed that smoking behavior is related to an increase in ERI at work, such as high demand, low control and low return [31]. Silveira et al. also found that the work stress level of industrial workers has a great impact on smoking [32]. Like smoking, the exciting effect of ethanol also makes alcohol drinking another way for people to reduce psychological tension. Vidal et al. found that offshore oil workers exposed to workplace stress had higher odds of alcohol abuse when compared to unexposed workers [33]. Occupational stress factors are related to an increase in alcohol consumption [34]. Research also showed that there is a close relationship between high work stress and obesity, and high work stress is a risk factor for increasing obesity [35]. The lower the level of job control, the higher the level of occupational stress, and the higher the degree of ERI, the higher the BMI. The increase in BMI may be caused by overeating caused by occupational stress, but physiologically, mental stress causes endocrine disorders and fat redistribution, which is also a reason for the increase in obesity. In order to eliminate tension at work, employees may turn to smoking, alcohol drinking and excessive eating to relieve the pressure. Therefore, the prevalence of hypertension is significantly increased when smoking, alcohol drinking or obesity are combined with occupational stress. This is of great significance for determining populations at high risk of hypertension and preventing and treating hypertension. Occupational stress is the main social and psychological stress factor faced by occupational groups in China. Long-term occupational stress can lead to physical or mental health problems and even related diseases. It is suggested to carry out health education and health promotion for petrochemical employees, make them aware of their behavioral characteristics, strengthen their psychological control ability and establish a good interpersonal system so as to alleviate tension and actively change unhealthy behaviors. Smoking and alcohol-drinking behaviors should be included as important subjects of health education, and the moral awareness education of employees should be strengthened.

## 5. Conclusions

Our findings reveal that there may be a high prevalence of hypertension in Chinese petrochemical workers. Demographic factors, work conditions and occupational stress were related to the development of hypertension. These factors not only have independent effects on hypertension in petrochemical workers but also have positive interactions in petrochemical workers. We will carry out health education and regularly organize lectures on hypertension. Setting up a psychological consultation room to relieve the psychological pressure of workers and reduce occupational stress would be beneficial. Interventions aiming at risk factors can effectively prevent the dangerous non-reversible consequences of hypertension.

## Figures and Tables

**Table 1 ijerph-19-16932-t001:** Characteristics of the classified subjects.

Variables	Total (%) *n* = 1488	Hypertension (%) *n* = 511 (34.3%)	Non-Hypertension (%) *n* = 977 (65.7%)	χ2	*p*
Age (years)				78.050	<0.001
≤35	369 (24.8)	79 (5.3)	290 (19.5)		
36–45	411 (27.6)	116 (7.8)	295 (19.8)		
46–55	598 (40.2)	252 (16.9)	346 (23.3)		
≥56	110 (7.4)	64 (4.3)	46 (3.1)		
Gender				15.739	<0.001
Female	352 (23.7)	90 (6.0)	262 (17.7)		
Male	1136 (76.3)	421 (28.3)	715 (48.0)		
Marital status				12.814	<0.001
Single/divorced	239 (16.1)	58 (3.9)	181 (12.2)		
Married/cohabiting	1249 (83.9)	453 (30.4)	796 (53.5)		
Educational level				58.985	<0.001
Junior college or below	762 (51.2)	332 (22.3)	430 (28.9)		
Senior high school or above	726 (48.8)	179 (12.0)	547 (36.8)		
BMI				31.715	<0.001
<25	901 (60.6)	259 (17.4)	642 (43.2)		
≥25	587 (39.4)	252 (16.9)	335 (22.5)		
Smoking				37.634	<0.001
No	982 (66.1)	284 (19.1)	698 (47.0)		
Yes	506 (33.9)	227 (15.3)	279 (18.6)		
Alcohol drinking				22.452	<0.001
No	547 (36.8)	146 (28.6)	401 (41.0)		
Yes	941 (63.2)	365 (71.4)	576 (59.0)		
Physical exercise status				3.886	0.143
Never	286 (19.2)	92 (6.2)	194 (13.0)		
Occasional	667 (44.8)	218 (14.7)	449 (30.1)		
Often	535 (36.0)	201 (13.5)	334 (22.5)		
Monthly income (CNY)				2.743	0.098
≤4000	862 (57.9)	311 (20.9)	551 (37.0)		
>4000	626 (42.1)	200 (13.4)	426 (28.7)		
Job rank				14.739	<0.001
Head workers	320 (21.5)	81 (5.4)	239 (16.1)		
Staff workers	1168 (78.5)	430 (28.9)	738 (49.6)		
Weekly work time (h/week)				5.073	0.024
≤40	906 (60.9)	291 (19.6)	615 (41.3)		
>40	582 (39.1)	220 (14.8)	362 (24.3)		
Occupational category				21.478	<0.001
Refinery	573 (38.5)	161 (10.8)	412 (27.7)		
Petrochemical	237 (15.9)	87 (5.8)	150 (10.1)		
Storage and transportation	292 (19.7)	127 (8.5)	165 (11.2)		
Others	386 (25.9)	136 (9.1)	250 (16.8)		
Work shift				1.200	0.273
No	830 (55.8)	295 (19.8)	535 (36.0)		
Yes	658 (44.2)	216 (14.5)	442 (29.7)		
Night shift				0.311	0.577
No	515 (34.6)	172 (11.6)	343 (23.0)		
Yes	973 (65.4)	339 (22.8)	634 (42.6)		
Occupational stress					
ERI				8.344	0.004
≤1	1167 (78.4)	379 (25.4)	788 (53.0)		
>1	321 (21.6)	132 (8.9)	189 (12.7)		

**Table 2 ijerph-19-16932-t002:** Multivariable logistic regression on influencing factors for prevalence of hypertension.

Variables	B	S.E.	Wald χ^2^	*p*	OR	95%CI
Age (years)						
≤35						
36–45	0.113	0.198	0.326	0.568	1.120	0.759–1.652
46–55	0.653	0.197	11.016	0.001	1.922	1.307–2.826
≥56	1.143	0.280	16.530	<0.001	3.190	1.841–5.526
Gender						
Female						
Male	0.362	0.157	5.323	0.021	1.436	1.056–1.954
Marital status						
Married/cohabiting						
Single/separated	0.322	0.175	3.386	0.060	1.391	0.979–1.946
Educational level						
Junior college or below						
Senior high school or above	−0.554	0.139	15.918	<0.001	0.574	0.438–0.754
Physical exercise status						
Never						
Occasional	−0.018	0.162	0.012	0.912	0.982	0.715–1.350
Often	0.098	0.168	0.336	0.562	1.102	0.793–1.533
Monthly income (CNY)						
≤4000						
>4000	−0.195	0.131	2.214	0.137	0.825	0.638–1.065
BMI						
<25						
≥25	0.623	0.120	26.734	<0.001	1.856	1.468–2.346
Smoking						
No						
Yes	0.421	0.126	11.203	0.001	1.524	1.191–1.951
Alcohol drinking						
No						
Yes	0.422	0.131	10.401	0.001	1.525	1.180–1.972
Job rank						
Head workers						
Staff workers	0.224	0.168	1.775	0.183	1.251	0.905–1.749
Weekly work time (h/week)						
≤40						
>40	0.047	0.121	0.149	0.700	1.048	0.827–1.740
Occupational category						
Refinery						
Petrochemical	0.209	0.176	1.419	0.234	1.233	0.874–1.740
Storage and transportation	−0.018	0.186	0.009	0.924	0.982	0.682–1.416
Others	0.189	0.157	1.448	0.291	1.208	0.888–1.642
ERI						
≤1						
>1	0.399	0.140	8.152	0.004	1.490	1.133–1.960

**Table 3 ijerph-19-16932-t003:** Analysis of interaction between occupational stress and risk factors on hypertension.

Variables		Hypertension (*n* = 511)	Non-Hypertension (*n* = 977)	OR1/RR1 (95%CI)
Smoking	ERI			
−	−	212	562	1 (reference)
−	+	72	136	3.001 (1.881–4.788)
+	−	167	226	2.361 (1.291–4.318)
+	+	60	53	3.184 (2.008–4.485)
OR2/RR2 (95%CI)				6.496 (3.883–10.867)
RERI				2.134 ^a^ (0.085–4.182)
AP				0.318 ^b^ (0.083–0.574)
S				1.635 ^c^ (1.021–2.616)
Alcohol drinking	ERI			
−	−	115	323	1 (reference)
−	+	31	78	2.995 (1.933–4.639)
+	−	264	465	2.130 (1.183–3.835)
+	+	101	111	2.324 (1.813–3.603)
OR2/RR2 (95%CI)				6.457 (3.860–10.802)
RERI				2.332 ^a^ (0.264–4.401)
AP				0.361 ^b^ (0.127–0.595)
S				1.746 ^c^ (1.083–2.815)
BMI ≥ 25	ERI			
−	−	192	521	1 (reference)
−	+	67	121	2.685 (1.819–3.962)
+	−	187	267	1.882 (1.046–3.389)
+	+	65	68	2.698 (1.777–3.786)
OR2/RR2 (95%CI)				5.408 (3.205–9.125)
RERI				1.841 ^a^ (0.590–3.093)
AP				0.339 ^b^ (0.193–0.488)
S				1.717 ^c^ (1.243–2.372)

**Notes:** OR: odds ratio; RR: Risk Ratio; OR1/RR1: multiplicative interaction; OR2/RR2: additive interaction; CI: confidence interval; RERI: relative excess risk of interaction; S: synergy index; ^a^: RERI > 0 means a positive interaction or more than additive; ^b^: AP > 0 means a positive interaction or more than additive; ^c^: S > 1 means a positive interaction or more than additive. Adjusted variables: age, gender and educational level.

## Data Availability

The datasets generated and/or analyzed during the current study are not publicly available because the data form part of an ongoing study but are available from the corresponding author on reasonable request.

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
