# Peer review of "The Interaction between Occupational Stress and Smoking, Alcohol Drinking and BMI on Hypertension in Chinese Petrochemical Workers"

_ijerph, 2022, doi:10.3390/ijerph192416932_

Round 1
Reviewer 1 Report
The presented study explored the impact of the interaction between occupational stress and smoking, alcohol drinking, and BMI on hypertension in Chinese petrochemical workers. A total of 1900 employees participated, including employees who have worked for more than one year.
The Authors confirmed well-known information that older age, male gender, BMI≥25, lower education, smoking, and alcohol drinking are risk factors for hypertension. The Authors confirmed that occupational stress interacts with smoking, drinking, and obesity to affect hypertension.
Remarks:
1. Table 1 – the percentage of persons with hypertension in relation to the given subpopulation would be more informative, than the structure of hypertensive and nonhypertensive persons by a given characteristic.
2. It would important to know how many of the people with BP≥140 mmHg and/or a DBP≥90 mmHg were defined as having hypertension, had a possibility for clinical observation in occupational/ family outpatient clinics, and how many such cases were confirmed.
3. Whenever in-text “drinking” is mentioned should be replaced by “ alcohol drinking:
Author Response
Thank you for your valuable suggestions. We have made the following modifications according to your suggestions.
Response1: Thank you. We have modified Table 1 according to your suggestions.
Response2: In our study, 398 workers had SBP≥140 mmHg and DBP≥90 mmHg; 56workers only had SBP≥140 mmHg; 39 workers only had DBP≥90 mmHg; 18 workers whose value of BP was normal, but they took anti-hypertensive drugs for a long time, and the medical staff also classified them as hypertension patients according to their symptom descriptions.We have supplemented in Section 2.2 of the manuscript in Line 125-130.
The 511 hypertensive patients in our study were all in mild to moderate hypertension, and there were no patients with severe hypertension requiring clinical observation.
Response3: We changed all “drinking” in the manuscript to “alcohol drinking” and modified them in the manuscript.

Reviewer 2 Report
Dear Authors,
I read with very pleasure your work, which is really interesting and well written. I only have few suggestions, particularly for the methods section.
line 35: what survey?
line 74: don't you believe that the type of job itself (hard, strenous like that in petrochemical plants) can lead to a poor lifestyle? I suggest to introduce this aspect also in introduction.
Where did the participants come from? All China? All petrochemicals industry? How did you select the 1900 possible participants?
Phases of the study are not clear. Where the blood measuraments took place? When (year, month)?
"Exclusion criteria: workers with family history of mental illness and/or hy- 95 pertension; workers with clear secondary hypertension.", then in the paragraph below you stated: "Workers who used antihypertensive medications or were considered 107 hypertension by doctors who worked in a formal medical institution were also classified 108 as hypertensive, regardless of the measured levels of blood pressure". So, are they included or excluded?
The fact that you have in your smaple both blue and white collars (head and staff workers...am I correct? you should provide som insight about the type of job, I presume that head workers conduct intellectual activities and staff manual activities, but you should clarify it), should be addressed in discussion, since finding educational level protective can be related to the type of job performed.
Author Response
Thank you for your detailed suggestions on my method, which is very helpful to me. We have made the following modifications according to your suggestions.
Response1: The survey was about the prevalence of hypertension among adults in northeast China. We have supplemented in the manuscript in Line37-39.
Response2: Yes, I believe that. I have made a corresponding supplement in the introduction in Line 70-77, 83-84.
Response3: A cross sectional survey was conducted in Panjin City, Liaoning Province, China in May 2019. Liaoning Province is an important heavy industry base in China, and Panjin City gathered a lot of petrochemical enterprises. Finally, we randomly selected a large factory, and a total of 1900 workers were willing to participate. We have supplemented in Section 2.1 in the manuscript in Line100-103.
Response4: A cross sectional survey was conducted in Panjin City, Liaoning Province, China in May 2019. We have supplemented in Section 2.1 in the manuscript in Line 100-102.
The place to fill in the questionnaire and measure BP was in the quiet office of the factory. Investigators were divided into four groups to carry out the investigation at the same time. The investigation lasted for 10 days, and about 200 people were investigated every day. BP measurement was the most accurate in the morning when the participants were on an empty stomach, so it started at 8:00 every morning. Before filling in the questionnaire, trained nurses used BP monitors to measure the BP of workers. We have supplemented in Section 2.2 in the manuscript in Line 113-118.
Response5: They were included. In our study, there were 18 workers whose value of BP was normal, but they took anti-hypertensive drugs for a long time, and the medical staff also classified them as hypertension patients according to their symptom descriptions. We have supplemented and modified this description in Section 2.2 of the manuscript in Line 128-130.
These 18 workers were not secondary hypertension. It might be that they took antihypertensive drugs to control their blood pressure, so that the blood pressure was normal when we measured it. However, despite this, the medical staff thought that these workers were suffering from hypertension, so they were still classified as hypertension.
Response6: Thank you, you're right. Our research objects include both blue and white collars (head and staff workers. I added relevant insight in the discussion in Line245-251.
